# Construction of Inorganic Bulks through Coalescence of Particle Precursors

**DOI:** 10.3390/nano11010241

**Published:** 2021-01-18

**Authors:** Zhao Mu, Ruikang Tang, Zhaoming Liu

**Affiliations:** 1Department of Chemistry, Zhejiang University, Hangzhou 310027, China; zhaomu@zju.edu.cn (Z.M.); rtang@zju.edu.cn (R.T.); 2State Key Laboratory of Silicon Materials, Zhejiang University, Hangzhou 310027, China

**Keywords:** inorganic material, particle, precursor, coalescence, bulk materials

## Abstract

Bulk inorganic materials play important roles in human society, and their construction is commonly achieved by the coalescence of inorganic nano- or micro-sized particles. Understanding the coalescence process promotes the elimination of particle interfaces, leading to continuous bulk phases with improved functions. In this review, we mainly focus on the coalescence of ceramic and metal materials for bulk construction. The basic knowledge of coalescent mechanism on inorganic materials is briefly introduced. Then, the properties of the inorganic precursors, which determine the coalescent behaviors of inorganic phases, are discussed from the views of particle interface, size, crystallinity, and orientation. The relationships between fundamental discoveries and industrial applications are emphasized. Based upon the understandings, the applications of inorganic bulk materials produced by the coalescence of their particle precursors are further presented. In conclusion, the challenges of particle coalescence for bulk material construction are presented, and the connection between recent fundamental findings and industrial applications is highlighted, aiming to provide an insightful outlook for the future development of functional inorganic materials.

## 1. Introduction

Bulk inorganic materials are essential in humans’ daily life, including building construction, optical devices, biomaterials, and mechanical materials [1,2,3]. An advanced technique for the construction of bulk-scaled materials is necessary to ensure their applications. Currently, the construction of these materials is commonly achieved by the compaction of their particle-precursors [4,5,6,7]. It is mainly because crystallization is the most common tactic for the production of most inorganic materials, resulting in nano- or micro- sized solid powders. The discontinuous particle interfaces and boundaries may result in defects, thus materials with continuous structures can always have superior performance to that of materials with defects [8,9,10,11]. It can be simply understood that, when a material with excellent mechanical property exists defects, external stress can easily destroy the material through the propagation of cracks [12,13]. Besides the limitation on the mechanical properties, discontinuity also reduces other performances of materials, such as light transmittance [14], conductivity [15], and thermal conductivity [16]. In order to improve the continuity of inorganic materials, the coalescence of particles is established so as to fabricate bulk materials from particle-precursors with improved continuity [17,18]. This review will focus on the coalescence of ceramic and metal materials. Understanding the influence factors on the coalescence of ceramic and metals particles from a fundamental view can fill the knowledge gaps to improve the technique for the construction of continuously structured inorganic bulk materials.

## 2. Basic Understanding of Particle Coalescence and Bulk Construction

Coalescence is a process in which two or more particles merge to form a single particle during contact. It is complex because many different stages or pathways may occur simultaneously during the coalescence [19]. According to the previous studies, the mechanism of nanoparticle coalescence is attributed to the mass transport, which contains four modes including surface diffusion, hydrodynamic flow, evaporation versus condensation, and volume diffusion [20]. McCarthy et al. conclude that the surface diffusion is recognized as the dominant mode [21]. However, Lewis et al. perform molecular dynamic simulation and indicate that, sometimes, surface diffusion may be overestimated during coalescence [22]. Panagiotis et al. try to decompose the process of coalescence into nine basic processes, including coagulation, rigid body reorientation, defect formation, heat release due to free-surface annihilation, temporary melting of interface, neck growth, plastic deformation, consolidation, and slow aging [19]. Luckily, the origin of coalescence, which has a clear recognition, is controlled by thermodynamics [23].

From a thermodynamic standpoint, coalescence is driven by a reduction of the surface energy [24]. A perfect model of the surface starts from an ideal crystal, which is shown in Figure 1a. Every atom has a particular repeating site and the chemical interaction between atoms is represented by lines. If these perfect crystals are broken, the atoms on the surface of the crystal lack coordination, leading to the generation of dangling atomic bonds, which induces surface energy (Figure 1b). The surface energy can be understood as the excess energy at the surface of a particle to the inner phase. It is related to the density of dangling bonds per unit area, and determined by the crystal facets and compositions. Thus, the existence of dangling bonds drives the attraction, aggregation, and coalescence between particles, so as to minimize surface energy [25].

For the particle fusion process, the interface energy caused by dangling bonds is essential, but in addition, the lattice matching between particles is also very important. There is a close relationship between the grain boundary energy and the angle, which is just introduced through a rough and qualitative discussion in Figure 1c,d. The grain boundary energy here can be understood as the energy barrier for particle coalescence. Typically, the grain boundary energy relates to the degree of misorientation between two particles. Because the grain boundary energy might be high at a facet with more unmatched bonds, or low at a facet with a good bonding matching, some researches establish the correlation between the grain boundary energy and the angle. In general, low angle boundaries can be analyzed by the Read–Shockley model (Figure 1d) [26], and a typical value of this “low angle” in cubic materials is around 15°. However, the energy evaluation for high angle boundaries is more complex, and no universal model can describe the relationship at present [27]. Notably, a very high angle will directly inhibit the fusion of particles owing to a very high grain boundary energy. Better lattice matching is more conducive to particle fusion. This part of the content, including the self-rotation behavior, will be explained in detail in the section on “crystallographic orientation”.

However, during the compaction of particles for bulk material construction, most situations are the compaction of many misoriented particles. Even though imperfect coalescence will lead to dislocations during the initial fusion, a further reconstruction may occur to orient the dislocations. In the experiment of electron beam induced coalescence between Au nanoparticles, the result indicates that initial dislocations can climb and slip to the particle surface to be removed through structural reconstruction [28]. In the process of coalescence between two Pd nanoparticles, the crystallization wave can be propagated and then transfer, triggering lattice reconstruction, and lead to a mono- or polycrystalline structure [29]. This phenomenon is also observed in ceramic materials such as calcium carbonate [30].

As different particles have different areas of solid–vapor or solid–solid interface, the driving force of coalescence may differ in the same process. The coalescence between particles is driven by the system thermodynamics towards the reduction of surface energy. However, the average energy gradient between solid–vapor surface and solid–solid interface is small, which is within the range of 1~2 J/m^2^ for most ceramic materials. The energy is insufficient to break most chemical bonds, inhibiting the free movement of atoms for particle coalescence. When the driving force for continued coalescence is diminished, the process will gradually slow as the surface energy is consumed. Thus, natural coalescence hardly occurs at room temperature or conditions. The heat stimulation is needed to accelerate the rate of coalescence. The temperature is the most important factor in the process of coalescence. During the heating period, the atoms in materials move faster along particle surfaces, along the grain boundary, and through the lattice during heating. The temperature not only accelerates coalescence, but also changes and controls the transport pathways of coalescence [25].

## 3. The Influence Factors on the Coalescence of Particle-Precursors

To develop alternative methods, understanding the basic influence factors on the coalescence of inorganic particles is necessary. In this section, we mainly summarize the influence of particle interface, size, crystallinity, and orientation on the coalescence of inorganic particles from a fundamental view, demonstrating some recent achievements as well as potential methods that benefit the construction of bulk materials.

### 3.1. Particle Interface

As the driving force of particles coalescence is the surface energy of particles, the particle interface is the most concerning factor for coalescence. In this part, we mainly introduce particle coalescence from the particle interface. The strategy is to promote the atomic transportation between particle boundaries. According to material characteristics, different methods have been developed to promote this process. The heating is the most extensive way to promote the atomic transportation between particle interfaces [18]. Besides heating, solvent-assisted methods, such as cold sintering [31], and additive-assisted interface coalescence [32] have also been developed according to material solubility in specific solvents and interface affinity, respectively.

#### 3.1.1. Thermal-Assisted Interface Coalescence

The thermal treatment is the most mature method to increase atomic motion on particle surface for coalescence. Many methods have been developed based on thermal treatment; most are based on the sintering technique. Recently, some novel methods, such as microwave-assisted sintering [33], electrical current-assisted sintering [34], and resistance sintering [35], have been established to improve the traditional sintering method.

##### Traditional Sintering

Traditional sintering, already having a long history, is the primary technique for the coalescence of particle-precursors for bulk ceramic material construction at a high temperature. The most important application of sintering is the production of ceramic materials. Ceramic material is a solid material comprising an inorganic compound of metal and non-metal with ionic or covalent bonds, which cause most ceramic materials to have good hardness and toughness and poor conductivity [36]. This technique is a thermal treatment for bonding ceramic particles into a coherent, predominantly solid structure via mass transportation that often occurs at the atomic scale [37]. Nowadays, traditional sintering is still the most primary and widest operation in the production of most ceramics—white wares, refractories, bricks, abrasives, porcelain, cutting tools, medical materials [38], and construction materials [39]. For example, silicon nitride ceramics exhibit unique properties owing to their high fracture toughness and super-plasticity, and they mainly act as structural materials for high-temperature application [40]. Sintering has mainly been used for the fabrication of fine-grained silicon nitride under hot pressing [41].

The traditional sintering protocol is very important, but it also has some drawbacks. One is that many ceramic compounds have strong covalent or ionic bonds, which need an extremely high temperature to initiate mass transportation for coalescence. Traditional sintering that uses thermal radiation for heating is less efficient to promote mass transportation, and in turn, becomes energy consuming [25]. Besides, many mineralogical ceramics are less stable under a high temperature, which limits the application of traditional sintering. Thus, it motivates researchers to improve the efficiency of the heating protocol on inorganic particles. Recent studies have tried utilizing the characterization of specific materials to develop different techniques for thermal-assisted particle coalescence, so as to promote the efficiency of the heating protocol. Many novel thermal-assisted methods, such as microwave-assisted sintering [33], electrical current-assisted sintering [34], and resistance sintering [35], have been developed to achieve this goal.

##### New Methods Based on Thermal-Assisted Interface Coalescence

The microwave sintering process has a unique advantage over conventional sintering processes. The difference lies in the heating mechanism. In conventional sintering, the heating element generates heat through thermal radiation. Unlike conventional heating, in microwave heating, energy is transferred to material through the interaction of the electromagnetic field at the molecular level. The interaction of microwaves in a dielectric material results in the rotation of the dipoles and translational motions of free or bound charges. The resistance of these induced motions due to frictional, inertia, and elastic forces results in volumetric heating. The materials themselves absorb and transform microwave energy into heat within the sample volume and coalescence can be completed in shorter times. The dielectric properties of materials determine the effect of the electromagnetic field on the material [33].

Resistance sintering is a kind of new sintering method that utilizes a high direct current or alternating current to heat conductive particles. The atomic motion in conductive particles is accelerated and the coalescence of particle interface can be achieved. Compared with conventional sintering, the heat is generated only inside the particle, and then transferred to the interface. Thus, resistance sintering displays more heating efficiency, achieving sintering in a short amount of time [35].

The spark plasma sintering (SPS) technique is another new method modified with the hot-pressing process. This technique also belongs to thermal-assisted coalescence and is similar to conventional sintering. The typical feature of the SPS process is the application of the current and higher heating rate [34]. This is different from conventional sintering in the way of generating heat. The powder is placed within a conducting metal or carbon die, then the die generates heat when a pulsed direct electric current is applied between dies. Generally, the applied electric field in SPS is below 10 V∙cm^−2^. This technique can also achieve a high-speed heating rate [42]. The SPS process is performed in low vacuum (almost 3 Pa) in order to avoid oxidation of the graphite die at high temperatures. This method had some achievements, for example, increasing the super plasticity of ceramics [42], higher permittivity in ferroelectrics [43], and reduced impurity segregation at the grain boundary [44].

Recently, Hu et al. developed a ceramic synthesis method, called the ultrafast high-temperature sintering (UHS) process. The ceramic materials are manufactured through radiative heating under an inert atmosphere. The UHS method is shown in Figure 2a. First, the raw materials are weighed according to the stoichiometric ratio, and then mixed and compressed. Then, the embryonic body piece is directly placed in the UHS heating device, the temperature is quickly raised to the target temperature and maintained for about 10 s, and then the temperature is lowered to room temperature to obtain a compact structure, which can exhibit excellent performance. Notably, the whole sintering process is about one minute. The researchers indicate that the sintering process is indeed very different from the traditional sintering method (Figure 2b). During the sintering process within 10 s, the solid-phase reaction and sintering process of the embryonic body sample proceed simultaneously. Unlike the traditional sintering method, which clearly divides the initial, middle, and late stages, there is no obvious stage distinction for the changes in sample porosity and grain size during UHS sintering. The rapid sintering process of UHS can effectively inhibit the loss of volatile elements in the ceramic components (Figure 2c); a high sintering temperature can ensure the compactness of the ceramic structure, and the density of the synthesized ceramic is as high as 94%. UHS sintering can be combined with structural design to produce layered composition material, for three-dimensional (3D) printing. It also has potential applications in solid-state electrolytes, multicomponent structures, and high-throughput materials screening (Figure 2d–g) [45].

#### 3.1.2. Solvent-Assisted Interface Coalescence

Although thermal-assisted interface coalescence has been widely used in the production of materials, there are still some limitations. As thermal-assisted interface coalescence is mainly driven by the thermodynamics, the rate of sintering is mostly dependent on temperature and, typically, the atomic motion is promoted with the increase of temperature [25]. Conventional sintering of polycrystalline ceramics is usually performed at quite a high temperature, normally up to 1000 °C for most ceramic materials, typically 50% to 75% of their melting temperatures [46]. In some cases, high temperatures are restrictive in the context of material integration, material synthesis, and phase stability, which limit the scope of application. Thus, it is necessary to develop alternative methods to promote the coalescence of particle-precursors for bulk material construction.

The key for particle coalescence is to improve the mass transportation within grain boundaries. Except for thermal-induced atomic motion, the atoms dispersed in solvents can also have excellent diffusivity. The atom/ions have almost higher coefficient of diffusion in solution than in solid [47]. Thus, for the dissolved inorganic materials, the mass transportation between particles can be realized by adding suitable solvents at boundaries, promoting the coalescence of particles. Utilizing the solubility of the particle surface, a new coalescent method at a low temperature was created. Because of the relative low sintering temperature, this approach is named the “cold sintering process” (CSP). During cold sintering, a small volume fraction of solvents is firstly mixed with initial powders. Solvent can be chosen according to the solubility of the material in the solvent to cause moderate dissolution. Water is usually the most common solvent in cold sintering, while other organic and ionic liquids are also used. Simultaneously, pressure is applied to improve particle densification [48]. The process of cold sintering is usually understood as the dissolution of materials in cold sintering solvent, and the dissolved atoms/ions will move through solvent under pressure. When a balance is achieved, pores are filled with re-precipitated atoms/ions (Figure 3a) [46].

Lots of inorganic materials use cold sintering to build bulk materials, including solid-state electrolytes [46], ferroelectrics [49], piezoelectric materials [50], Li-ion cathodes [51], structural materials [52], microwave dielectrics [53], semiconductors [54], and magnetic ceramics materials [55]. In 1986, Yamasaki first published the work about densifying ceramic at temperatures below 200 °C through hydrothermal processing and isostatic pressing [56]. Florian et al. use nano-vaterite to construct bulk ceramics through cold sintering at room temperature, which has specific strength. Its strength is comparable to that of materials like stone and concrete [57]. In addition to the densification of inorganic material, many other properties are found in the materials fabricated by cold sintering. Some materials are usually unstable at the high temperature of traditional sintering, so it is impossible to integrate them with ceramic powders through conventional sintering. Cold sintering achieves the combination of the ceramic powders and materials not resistant to high temperatures to achieve integrated properties and functions [46,52].

Owing to vast differences in the typical sintering temperatures of ceramics and polymers. It is unable to achieve the co-sintering of ceramic and thermoplastic polymers. Guo et al. creatively applied cold sintering in the design of ceramic-polymer systems (Figure 3b–d) [54]. In this work, three examples of ceramic-polymer systems were selected: microwave dielectric Li_2_MoO_4_ (LM)–(-C_2_F_4_-)_n_ (PTFE), electrolyte Li_1.5_Al_0.5_Ge_1.5_(PO_4_)_3_–(-CH_2_CF_2_-)_x_[-CF_2_CF(CF_3_)-]_y_, and semiconductor V_2_O_5_–poly(3,4-ethylenedioxythiophene) polystyrene sulfonate composite to demonstrate a wide and universal application. This method effectively solves the contradiction between ceramics and thermoplastic polymers in sintering, and provides a simple route to integrate ceramics and polymers that are traditionally incompatible. Similarly, Amanda et al. utilize the CSP to fabricate the flexible-printable Li_2_MoO_4_ capacitor array structures on both Nickel foil and PET film (Figure 3e,f); this would be impossible with conventional processing methods, because the Nickel foil would oxidize in air at temperatures above 300 °C, and the PET film would thermally degrade at temperatures of 225 °C to 260 °C [52]. Besides, high-temperature sintering is detrimental to battery integration. Lithium loss and second phase formation. CSP showed the first steps toward the application to low-temperature sintering and fabrication of electrolytes materials and similar materials [51].

#### 3.1.3. Additive-Assisted Interface Coalescence

Recently, additive-assisted interface coalescence has attracted growing interest [31,32]. Cho et al. can controllably prepare nanomaterials with various shapes, such as straight nanowires; zigzag, helical, branched, and tapered nanowires; as well as single crystal nano-rings [58]. Cheng et al. use iodine as the additive to induce gold nanoparticles coalescence and assembly [32]. They firstly find that iodine chemisorption takes place on gold nanoparticles, and the van der Waals force is the driving force for the coalescence. Martínez et al. induce coalescence of Au nanoparticles to form nanowires by functionalization and reduction of glucosamine phosphate [59]. Additive is an effective way to accelerate the coalescence of nanoparticles. On the one hand, additives can adjust coalescence behaviors of nanoparticles by absorption on specific crystal planes to change surface energy [60]. On the other hand, coalescence can be achieved by triggering the chemical reaction of a surface-modified molecule [59].

#### 3.1.4. Conclusions

Interface plays the most important role in the coalescence of particles based on current techniques. Thermal-assisted interface coalescence has presented a wide variety of applications for ceramic material construction. Novel methods, such as microwave sintering and SPS, are developed to improve the thermal efficiency for particle coalescence. Moreover, considering the thermal-induced side effects on some inorganic materials, other strategies for particle coalescence without heating are developed, such as solvent- and additive-assisted interface coalescence. Developing these methods based upon the unique characterization of inorganic materials could be a feasible way to minimize the drawbacks of thermal-assisted interface coalescence.

### 3.2. Size Distribution

Particle size is an important factor that affects particle coalescence. One is that the melting point of a metal particle is recognized as size-dependent [61]. With the particle size decreasing, the melting points of particles will decrease. This means that the effect of size on coalescence translates into an effect of temperature by correlating the melting points of metal particles with their size to allow particles to coalesce at a lower temperature. There is a mathematical foundation to describe the correlation between particle size and melting temperature [62]:$$T_{mp}=T_{mb}\;\left(1-\frac{N}{2n}\right)$$
where *T_mp_* and *T_mb_* are the melting temperature of nanoparticle and bulk material, respectively. *N* represents the number of the surface atoms and *n* represents total number of atoms contained in nanoparticle. It can be clearly obtained from this formula that, the smaller the particle size, the lower the melting point because, as the particle size becomes smaller, the ratio of surface atoms to the total number of atoms becomes higher. Simultaneously, this conclusion has also been confirmed in an experiment and theoretical simulation. Zhang et al. use the scanning calorimetry technique to measure the melting point of indium nanoparticles with a radius of 2 nm. The result indicates that the melting point decreases by 110 K compared with the melting point of bulk indium material [63]. Li et al. use molecular dynamics to study the coalescence of tungsten nanoparticles with different sizes [64]. They find that the rate of coalescence of smaller particles is obviously faster than that of bigger particles. The particles with diameters of 2 nm have achieved complete coalescence within 10 ns, while the particle with a diameter of 6 nm still stayed “neck” (Figure 4a). This conclusion can also be proved from the change of distance between the centers of mass of two nanoparticles (Rd) (Figure 4b). Because of the decrease of the total surface area, the total surface energy is released, leading to the effect of temporary heating. From the evolution of temperature T during the solid-solid coalescence process (Figure 4c), the energy difference causes a temperature increase of 562 K for a particle with a diameter of 2 nm, while the temperature only increases by 60 K for a particle with a diameter of 6 nm. Thus, the coalescence of two 2 nm particles is more complete and faster. Thus, size has two aspects of influence on coalescence: (1) smaller particles have a lower melting point and can achieve coalescence at a lower temperature and (2) the released energy for smaller particles during coalescence can cause a greater temperature increase. It is suggested that smaller particles are easier to coalescence. Moreover, the coalescence of two different sized nanoparticles is different from that of same sized nanoparticles. The particle with a smaller size firstly reaches the melting point. For example, for a 3 and 5 nm sized Ta nanoparticle system, the smaller nanoparticle that has a lower melting temperature will fully melt to wet the surface of the 5 nm particles [65]. This can be used to produce core–shell morphology [66].

Another interesting phenomenon is that, when the mineralogical particles reach an ultra-small size (~1 nm), they will spontaneously aggregate and coalesce to relatively larger matters. This process is revealed in many biological or biomimetic systems, in which ultra-small inorganic blocks, named as clusters, can aggregate for inorganic mineral formation. Denis et al. firstly demonstrate that stable pre-nucleation clusters are formed in the dissolved calcium carbonate solution, and pre-nucleation clusters may play an important role in the formation of minerals [67,68,69]. Habraken et al. investigate these pre-nucleation clusters in a calcium phosphate system by in situ analyzing their structures and crystallization process [70]. They point out that pre-nucleation clusters for calcium phosphate are in fact an aggregation of solution ion-association complexes, the Ca_2_(HPO_4_)_3_^2−^. Gower et al. discover the polymer-induced liquid precursor (PILP) system [71], in which the clusters of mineral precursors are stabilized by polymers. A variety of minerals with complex morphologies and textures can be formed by the aggregation of clusters in PILP under the control of various organic templates [72,73,74,75]. It is possible that ultra-small inorganic blocks have many unbounded surface atoms, leading to the high reactivity for coalescence. When this reactivity is controllable, the coalescence of particles for bulk material construction can be realized using ultra-small sized precursors, rather than traditional nano- or micro- sized particles. Recently, Liu et al. establish the end-capping strategy to control the reactivity of ultra-small sized precursors [76], named as inorganic ionic oligomers (Figure 5). Triethylamine (TEA) is used to stabilize inorganic ionic oligomers (the calcium carbonate, CaCO_3_, for example) via a hydrogen bond, so that a large quantity of ionic oligomers can be obtained (Figure 5a,b). The evaporation of TEA can directly induce the controllable crosslinking of ionic oligomers, resulting in the formation of continuously structured inorganic bulk material (Figure 5c). Consequently, centimeter-sized calcium carbonate mineral can be obtained, and the CaCO_3_ materials can be moldable constructed into complex shapes and morphologies (Figure 5d). This method can even be extended to repair damaged single crystals to recover their smooth surface. Moreover, the repaired region has exactly the same crystalline phase and lattice orientation, similar to a pristine sample (Figure 5e). Upon using this strategy, more ionic compounds, such as calcium phosphate and cupric phosphate, can be produced by the crosslinking of their oligomer precursors. Typically, the calcium phosphate ionic oligomers can be used to repair tooth enamel [77] and other hybrid bulk materials [78].

Except for the improvement of particle surficial diffusion, the size is also an effective factor for the bulk inorganic material construction by coalescence of particle precursors. We believe this alternative technique has competitive potential to traditional sintering, as long as the size of particle precursors can be rationally controlled.

### 3.3. Crystallinity

The crystallinity degree is another factor that can influence the coalescence behavior of inorganic particles [30,65,79]. The crystalline particles have a well-ordered, periodic structure, which means the atoms within crystals have well-defined locations and chemical bonds. In contrast, the atomic structures in poorly crystalline or amorphous particles are less oriented, which weakens or decreases the chemical bonds between atoms. In other words, the diffusion of atoms in poorly crystalline particles is easier, and the coalescence is more feasible. E. Landi et al. have evaluated the coalescence behavior of hydroxyapatite powders with different crystallinity [79]. They found that a low crystallinity degree of the starting powder highly favors the coalescence process. Panagiotis et al. investigate the coalescence behavior of amorphous and crystalline tantalum nanoparticles. The result shows that an increased crystalline misorientation can cause a full fusion at a lower temperature [65]. Simultaneously, the sintering rate was also affected by the degree of misorientation. The study reports that, for two of the same sized Au nanoparticles (a pair of 3 nm or 4 nm nanoparticles) at 800 K, the coalescence rate of crystalline nanoparticles is much lower than that of amorphous ones with the same size. The coalescent rate of amorphous particles is more than two orders of magnitude greater than that of high crystalline order nanoparticles [65].

In nature, the advantages of amorphous particles for the coalescence of bulk materials are well exhibited in the biomineralization process. Many biological organisms can produce inorganic skeletons with a complex morphology (Figure 6a,b), such as coccolith [80], sea urchins [81], and coral [82]. Importantly, their mechanical properties are always superior to the artificial materials with similar components; one of the reasons is that they have continuous mineral structures. More and more evidence suggests that the organisms use amorphous particles as precursors to form these structures [82,83]. The amorphous precursors are suggested to coalesce on the surface of biominerals to form a continuous amorphous–crystal interface (Figure 6c) [84]. Subsequent solid-state phase transition leads to the crystallization of amorphous phase to crystals while preserving the morphology of their amorphous precursors (Figure 6d–g). Because of the unique moldable characteristics of the amorphous phases, the biominerals can even have complex shapes and morphologies [83]. Indeed, many laboratory experiments have also revealed that poorly crystalline inorganic particles have the ability to spontaneously coalesce to form larger particles (Figure 6h).

Unluckily, even though the biological system can readily use amorphous particles to generate marvelous minerals, artificial construction of bulk materials using amorphous particle precursors still exists many problems. One major reason is that amorphous particles, especially for many minerals, are prone to crystallization, which significantly increases the difficulty for processing.

### 3.4. Crystallographic Orientation

For the coalescence of crystalline particles, the crystallographic orientation is also an important influencing factor. The coalescence may be inhibited when two particles are closely attached, while two well-oriented particles can easily coalesce. The imperfect alignment within particle-packed materials will lead to grain boundaries [86,87], which will cause bad effects on their properties, such as mechanical properties [88,89], tribological properties [90], and electrical [15] and semiconducting properties [91]. The oriented attachment (OA) is the most important phenomenon in particle-based crystal growth [92], which demonstrates the importance influence of crystalline orientation on particle coalescence. Coalescence and alignment occur simultaneously during OA growth. Since Penn et al.’s first report of OA in 1998 [93], extensive efforts have been carried out over the past decade to systematically investigate it, from experiments to theoretical understanding. The development of the in situ liquid-cell transmission electron microscopy method can potentially provide new insights through direct observation to OA, which greatly advances our understanding [94]. Li et al. use high-resolution in situ liquid-cell transmission electron microscopy to directly observe OA of iron oxyhydroxide nanoparticles (Figure 7a–j). The results reveal that OA is accomplished by a sudden jump over less than 1 nm after a perfect lattice match by continuous particles rotation and interaction. It has been proposed that the driving force for OA is the reduction of the total surface energy [95]. The detailed understanding of the properties of the solid-solvent structure is also key to advancing our understanding of OA. The structure of water around nanoparticles can be affected by particle size and morphology [96]. Zhang et al. measure the forces between rutile nanocrystals with different relative orientation through a combination of atomic force microscopy with environmental transmission electron microscope (TEM) [97]. The result demonstrated that the attraction is strongly dependent on relative orientation at a distance of approximately one hydration layer. Meanwhile, OA is highly sensitive to the properties of electrolyte solution. A huge change can be caused by tiny changes of solution ionic strength [98] or pH [99], leading to nanoparticle assembly from OA to random attachment.

For the thermal-assisted sintering, the importance of crystal orientation is also observed. In situ heating studies on indium tin oxide (ITO) nanoparticles clearly indicate that particle coalescence is completely dependent on the matching of crystallographic orientations (Figure 7k–m). High-resolution transmission electron microscope (HRTEM) is used to post characterize the lattice orientation of typical sintering stages in an interrupted and rapidly cooled sintering experiment. A small particle is attached to a bigger one without matching orientation (Figure 7k). As the orientation of the particles did not match, despite the close contact between two particles, the small particles cannot merge with bigger ones until up to a temperature of 1120 K. Reducing the angle to a range, coalescence can be achieved, but a twin boundary will be built between two particles, which cannot be eliminated in the process of the sintering experiment (Figure 7l). Particles can only coalesce completely when the crystallographic orientation is perfectly aligned (Figure 7m) [86]. The formation of a twin boundary can also be affected by the particle sizes [87]. In one simulation study on the two-particle geometry (Figure 7n), it was found that smaller particles are easier to re-orientate in the larger angle, leading to realigning of their crystallographic orientation across the particle neck region without a twin boundary (Figure 7o). However, because of the effect of sterical hindrance, the realignment of crystallographic orientation through rotation in solid-state coalescence is harder than the situation in the simulation.

If the crystalline orientation can be well controlled, it is also an alternative method to improve particle coalescence. One of the examples is the meso-crystal system. The meso-crystal is a special type of colloidal solid consisting of nanocrystalline building blocks that are well aligned [100,101,102]. A remarkable example is the formation of sea urchin spine. Each spine consists of highly oriented Mg-calcite nanocrystals. The spine diffracts as a single crystal of calcite and yet fractures as powder [103]. Meso-crystal is formed via a non-classical crystallization processes mediated by thousands of nanocrystals covered by organic additives. The meso-crystals can be a meta-stable state, which can be further transformed into single crystals through block fusion (Figure 8a). Cho et al. introduce different surfactant molecules to align different crystal faces, leading to different shapes, and then coalesce to form a single crystal (Figure 8b) [58]. Wang et al. observe Co_3_O_4_ aggregations with orientation alignment, then they removed the organic molecule to trigger particle coalescence by heating (Figure 8c) [104]. Unfortunately, the reaction for meso-crystal has not been achieved at a large scale. As long as the oriented assembly of particle precursors is rationally controlled, this strategy can have more potential applications in the construction of bulk materials.

## 4. The Coalescence of Materials for Functional Applications

The coalescence of inorganic particles is widely applied in many fields to form useful objects such as electronic capacitors, automotive transmission gears, metal cutting tools, and medical implants [5]. Coalescence is crucial to the success of several inorganic engineering produces, including a broad range of most ceramics, cemented carbides, and several metals. This part will briefly introduce the application of bulk materials produced by the coalescence of particle precursors.

### 4.1. Piezoelectric Materials

Piezoelectric materials are materials that can convert mechanical energy into electric energy, and have been used in many fields, including timekeeping, microphones, radio antenna oscillators, speakers, and hydrophones. Coalescence of particle precursors is widely applied in the fabrication of piezoelectric materials [106,107]. The sintering process was found to be able to improve the coalescence behavior, leading to optimized piezoelectric properties [107,108,109]. The densification of Na_0.5_K_0.5_NbO_3_ lead-free piezoelectric materials was densified through SPS (Figure 9), the density of which was raised to >99 % of the theoretical density, leading to an excellent piezoelectric parameter (146 pC/N).

Among the piezoelectric materials, the ferroelectric materials are crystalline materials that exhibit spontaneous electrical polarizations switchable by an external electric field [110,111]. Their large-scaled materials can be produced through coalescence from particles to bulks. Ferroelectric materials were firstly discovered in 1920 in the form of bulk single crystals. Since then, a number of ferroelectric materials have been developed in the form of bulk single-crystal and bulk polycrystalline ceramics [112]. In the group of ferroelectric materials, there are four subcategories: perovskite group, pyrochlore group, tungsten- bronze group, and bismuth layer structure group. Among these ferroelectric materials, perovskite is the most important and widely studied one. Masato et al. use the conventional sintering method to successfully synthesize well coalesced lead-free perovskite material with an excellent quality factor and electromechanical coupling factor, which have a marked potential as one of the lead-free perovskite materials [113]. Meanwhile, ferroelectric materials can be quickly produced through the coalescence of particles. Zélia et al. use laser heating to eliminate the interface of particles stack, rapidly obtaining Bi_4_Ti_3_O_12_ (BIT) ceramics bulk with a relative density of 94 ± 2% in 5 min [114].

### 4.2. Mechanical Materials

Coalescence can significantly improve the mechanical property of materials. Hydroxyapatite widely exists in the body as important components of hard tissues, such as teeth and bone [115]. Currently, hydroxyapatite is also used in the biological scaffold to induce the repair of biological hard tissue [116,117]. However, pure hydroxyapatite particles cannot be directly used, which limits the application of hydroxyapatite. Thus, the fabrication of bulk hydroxyapatite with excellent mechanical properties is momentous. The basic issue is the coalescence of hydroxyapatite particles. Many investigations have been done to promote the progresses in this field. Nowadays, some bulk hydroxyapatite materials with excellent mechanical properties are obtained through many methods [117,118,119,120]. S.J. Kalita et al. selected different sintering additives to in assist coalescence of hydroxyapatite (HAP) powder, and then tested the hardness of sintered compacts with different additives (Figure 10a,b). The result showed that the failure strength of HAP improved. Besides this, the mechanical properties of inorganic composite materials are improved through the achievement of coalescence [121]. Furushima et al. achieved coalescence using the sintering technique to change the mechanical properties of WC-FeAl composites. They find that effective integration of WC nanocrystals and FeAl nanocrystals through two kinds of sintering methods can be achieved, leading to excellent mechanical properties [121]. Choi use plasma spray to achieve coalescence of ZrO_2_ and Y_2_O_3_, and investigate the effect of the processing time on mechanical and physical properties. The results show that, except for phase stability, other properties, including flexure strength, fracture toughness, elastic modulus, and microhardness, increased significantly [122].

Except for the application in conventional materials, coalescence can also be used in the production of ultrahard materials [123]. Ultrahard materials primarily comprise polycrystalline diamond and polycrystalline cubic boron nitride [123]. These materials are commercially useful because of their unique combination of physical properties, which result in extremely high wear resistance combined with reliable mechanical behavior. Ultra-hard materials are primarily applied in some extremely demanding conditions, such as rock drilling as well as cutting [124]. Recently, Hitoshi et al. produce novel ultra-hard materials using nano polycrystalline diamond and nano polycrystalline cubic boron nitride through direct coalescence under ultra-high pressure and high temperature [125]. New hard materials do not contain any binder materials or secondary phases, so they have high strength compared with those of conventional sintered compacts containing binder materials. These novel hard materials are promising for applications to next-generation high-precision and high-efficiency cutting tools [39,126].

### 4.3. Medical Materials

The bulk materials are also widely used in medical fields including implants and dental inlay [4,31,127]. Titanium and its alloys have been used in medical materials as they have good biocompatibility and mechanical properties [128]. Laser sintering is a flexible way to achieve coalescence of particles, which can be combined with a virtual model to fabricate implant scaffolds with a complex structure according to individual patient data. Scaffolds with different shapes were manufactured through selective laser sintering (SLS) (Figure 10c–f). Three-dimensional printing was accomplished by SLS of titanium alloys powders to design and fabricate craniofacial implants with required shape and sizes [5]. Dental porcelain has also been successfully used in dental implants to construct dental restorations with high biocompatibility, high chemical stability, and resistance. In order to construct dental porcelain, it is necessary to mix the porcelain powder with solvent for shape, then heat treatment is applied to cause coalescence of particles [129]. Recently, Marina et al. consolidated amorphous calcium phosphate (ACP) by SPS. Consolidated ACP compounds have high meta-stability, allowing the fast release of bioactive ions upon resorption [31].

### 4.4. Optical and Functional Materials

Bulk optical materials are widely used in the fields of telecommunications, healthcare, energy production, and environmental monitoring [130]. To develop an economic way to prepare optical materials, the fabrication of optical ceramic material from nanoparticles has been suggested [131], and many ceramic optical materials have been developed. Hreniak et al. study the coalescence parameters of Nd:YAG nanoceramics, such as temperature and pressure. They find that the intensity of emission transitions is much higher than that of a single crystal and decreases with the increasing of pressure [131]. Ceramic converters have more thermal stability of high-power white light-emitting diodes, thus they can potentially replace traditional luminous silicone or resins. However, current ceramic converters lose enormous amounts of phosphor-converter light. Recently, Huang et al. prepared luminous hydroxyapatite-YAG:Ce ceramics at a high temperature by introducing bioceramic hydroxyapatite. High transparency for the first time is achieved in white light-emitting diodes equipped with phosphor in ceramics (Figure 11) [7]. Spinel, which can transmit electromagnetic and radiation in the visible and IR portion of the spectrum, shows promising applications in advanced electromagnetic windows and transparent armor. Gilde et al. compared the advantages and limitations of different methods [130]. Cubic yttria-stabilized zirconia has potential applications in solid electrolyte and oxygen sensors. Recently, its unique combination of mechanical and optical properties could extend its field of applications. Zhang et al. improved transparency through elaborating to investigate the sintering temperatures and demonstrated that the optimum sintering temperature is 1100 °C. The optimum transparency is obtained at this temperature [6].

## 5. Perspective

Coalescence is an important approach to manufacture inorganic bulk materials. It has a broad range of application in the fabrication of functional materials. Besides the traditional sintering technique, numerous novel methods have also been developed to promote particle coalescence for bulk material construction, such as cold sintering, ultrafast high-temperature sintering, SPS, and microwave sintering. With a deepened understanding of particle characters, such as interface, size, crystallinity, and orientation, many potential methods for bulk construction have been demonstrated in this review. A few discussions and perspectives are suggested to develop these methods for bulk inorganic materials. However, there are still several challenges.

Firstly, the current methods that are used to eliminate the particle interface within bulks are mainly driven by thermal assistance. Some novel methods, such as microwave sintering and SPS, are also based on thermal-induced coalescence. Cold sintering is a unique method that utilizes the dissolution-re-precipitate behavior of particles at the surface. However, a complete coalescence is difficult to obtain through this method. This issue also exists in thermal-assisted methods, unless the particles turn to liquid-like status at the melting point. However, it seems like a contradiction because the degradation temperature is often lower than the melting point for many inorganic materials. Exploring a balanced method that maintains the phase stability for inorganic materials and optimizes the particle coalescence is still the challenge.

Secondly, particle size plays a vital role in the conventional coalescence of particles. On the one hand, the melting point can obviously decrease with the reduction of the particles’ size. On the other hand, smaller particles have a higher surface energy, which benefits the coalescence of particles. Recently, Tang’s group synthesized ultra-small oligomers, which can control the coalescence of inorganic precursors just like polymerization [76]. This opens a new way to fabricate the inorganic bulk materials from the viewpoint of size. Perhaps, more attention can be paid to this influencing factor in the future for bulk material construction.

Moreover, inspired by biomineralization phenomena, an alternative strategy for manufacturing of monolithic inorganic materials is believed to be achieved by fusing their amorphous precursors. This could open the door in the synthesis of unique new materials. However, many attempts using amorphous particles under pressure have failed as a result of phase-transition-based crystallization rather than expected particle–particle fusion. The reason is that amorphous mineral phases such as amorphous calcium carbonate (ACC) and ACP are always rich in water. How to control amorphous particles to achieve the manufacture of monolith as biomineralization requires further in-depth research.

Developing a method to rationally control the particle orientation can also be feasible to build bulk inorganic materials. The phenomenon of OA has proven the feasibility from a thermodynamic view. The formation of mesocrystal further provides a potential strategy to control the orientation of inorganic particles. More fundamental studies could be done to have a comprehensive understanding of the self-orientation mechanism of particles, which may open up a novel technique for bulk material construction.

The construction of inorganic bulk material from particle precursors has made great progress from method to theory, but there are still many challenges and opportunities. We believe more fundamental mechanisms will be revealed through the understanding of particle coalescence, shedding light on more advanced functional bulk materials in the future.

## Figures and Tables

**Figure 1 nanomaterials-11-00241-f001:**
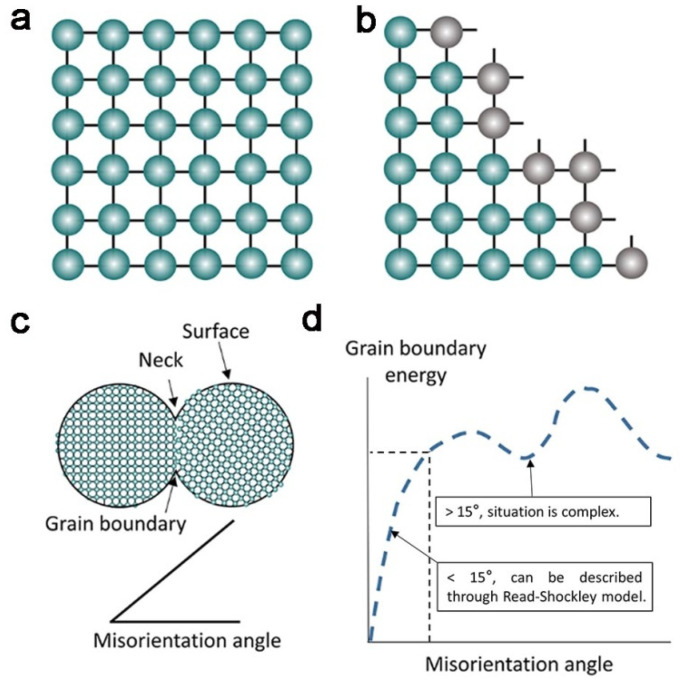
(**a**) Scheme of a single crystal structure, in which the atoms are in a repeating position, and the atomic bonds are linked between atoms. (**b**) Scheme of a free surface of a single crystal, showing disrupted atomic bonds. These dangling atomic bonds induce the surface energy; (**c**) scheme of two misoriented nanoparticles and the illustration of the misorientation angle; and (**d**) grain boundary energy changes with misorientation angle [25].

**Figure 2 nanomaterials-11-00241-f002:**
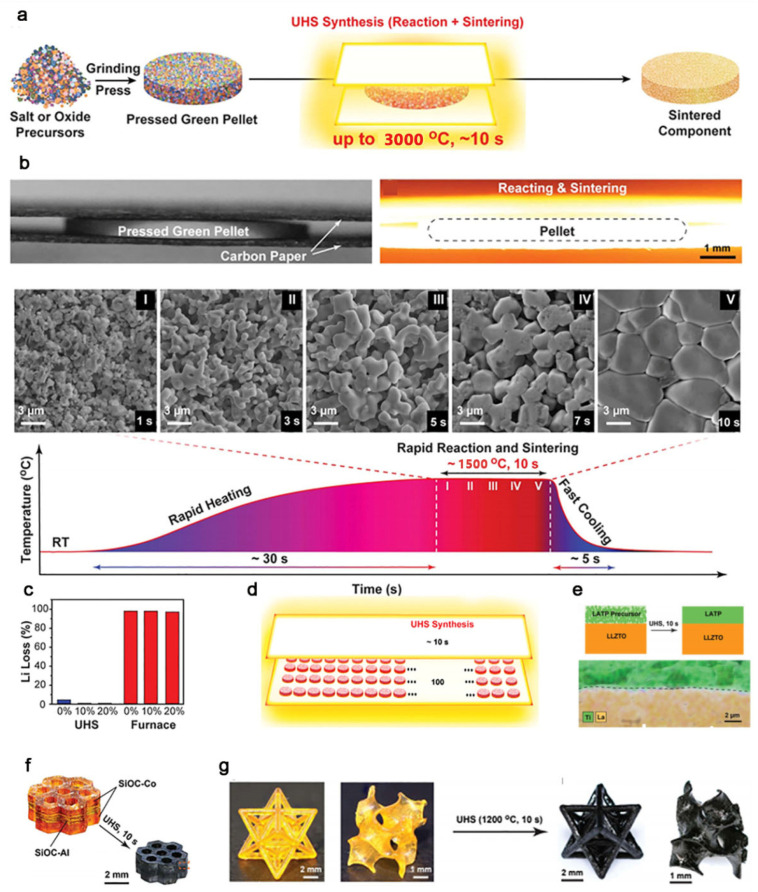
(**a**) Scheme for the protocol of the rapid sintering process; (**b**) the temperature profile of the ultrafast high-temperature sintering (UHS) process; (**c**) comparison of Li loss of different sintered Li_6.5_La_3_Zr_1.5_Ta_0.5_O_12_ (LLZTO) samples (0, 10, and 20% excess Li) using the UHS technique and conventional furnaces; (**d**) scheme of high-throughput screening: co-sintering ~100 matrix with ~10 s; (**e**) schematics and energy dispersive spectroscopy mapping of the co-sintered Li_1.3_Al_0.3_Ti_1.7_(PO_4_)_3_ (LATP)-LLZTO bilayer SSE; (**f**) a multilayer 3D-printed SiOC polymer precursor and a corresponding UHS-sintered sample; and (**g**) snapshots of the SiOC polymer precursor and SiOC samples sintered by the UHS method. Reproduced with permission [45]. Copyright 2020, AAAS. RT, room temperature.

**Figure 3 nanomaterials-11-00241-f003:**
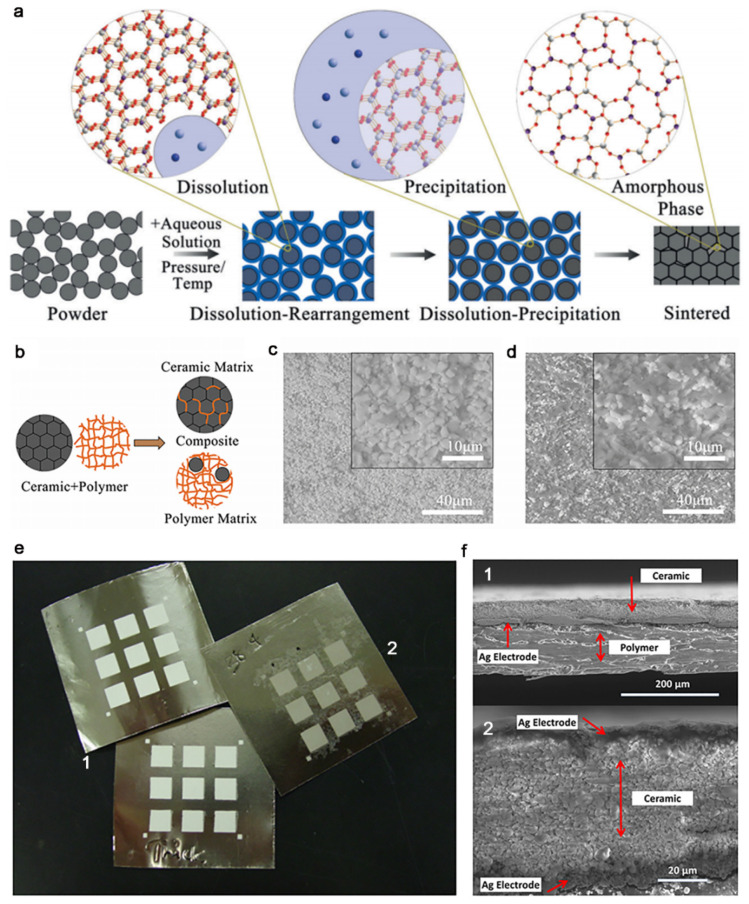
(**a**) Scheme of the cold sintering from particle precursors. Molecular structures at the grain boundary have been magnified. Reproduced with permission [46]. Copyright 2016, Wiley-VCH. (**b**) Scheme of cold co-sintered ceramic-polymer composites. (**c**,**d**) Scanning electron microscope (SEM) image of 90LM–10PTFE (**c**) and 40LM–60PTFE (**d**) composites, which are cold co-sintered at 120 °C and 350 MPa for 20 min. The white region is the Li_2_MoO_4_
_(_LM) phase, while the black region is PTFE phase. Reproduced with permission [54]. Copyright 2016, Wiley-VCH. (**e**) Capacitor arrays on Nickel foils by printing (1) or cold sintering (2). (**f**) SEM image on cross-sectional view of a cold-sintered Li_2_MoO_4_, showing a single-layered capacitor structure: (1) cold-sintered structure on polyethylene terephthalate film at low-magnification and (2) top and bottom electrodes in high-magnification image. Reproduced with permission [52]. Copyright 2016, The American Ceramic Society.

**Figure 4 nanomaterials-11-00241-f004:**
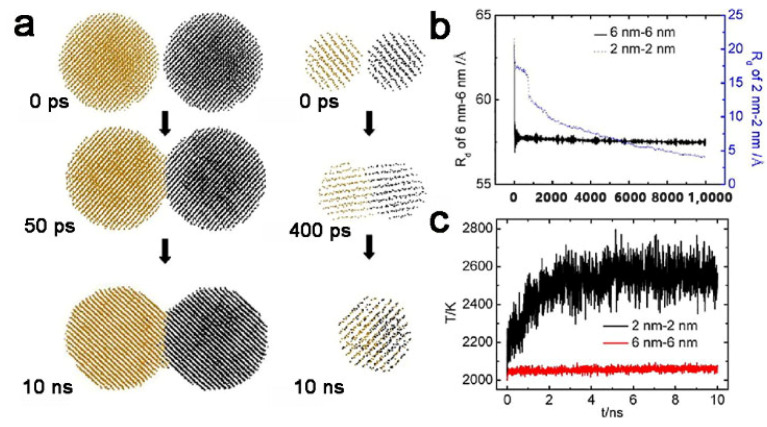
(**a**) Simulated structures of two solid nanoparticles during coalescence. Left column: two 6 nm nanoparticles; right column: two 2 nm nanoparticles. (**b**) The evolution of R_d_ diameters at 2000 K for solid-solid coalescence model. (**c**) The evolution of temperature T during solidsolid coalescence process. Reproduced with permission [64]. Copyright 2017, Elsevier B.V.

**Figure 5 nanomaterials-11-00241-f005:**
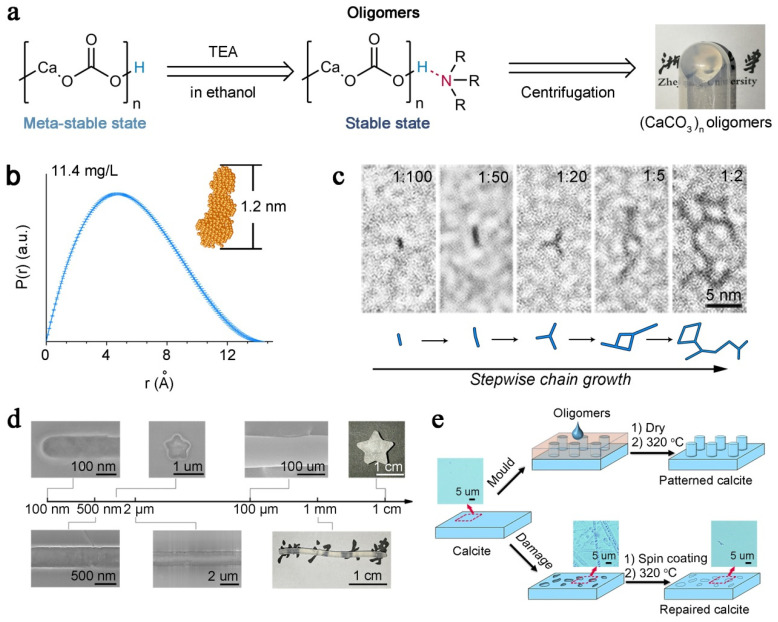
(**a**) Scheme of the capping strategy and reaction conditions for producing (CaCO_3_)_n_ oligomers; a photograph of gel-like (CaCO_3_)_n_ oligomers is presented on the right. (**b**) Pair–distance distribution function (P(r)) of the (CaCO_3_)_n_ oligomers. The shape simulation of the oligomer is shown in the inset. Error bars represent one standard deviation, n = 20. (**c**) high-resolution transmission electron microscope (HRTEM) images of (CaCO_3_)_n_ oligomers grown at different Ca/triethylamine (TEA) ratios from 1:100 to 1:2. (**d**) Molded CaCO_3_ with different dimensions and morphologies. (**e**) Schemes for pattern construction on single-crystalline calcite (top path), and the repair of rough single-crystalline calcite to smooth calcite (bottom path). The insets show optical microscopy images of the calcite surface at different stages: native, corroded, and repaired. Reproduced with permission [76]. Copyright 2019, Springer Nature.

**Figure 6 nanomaterials-11-00241-f006:**
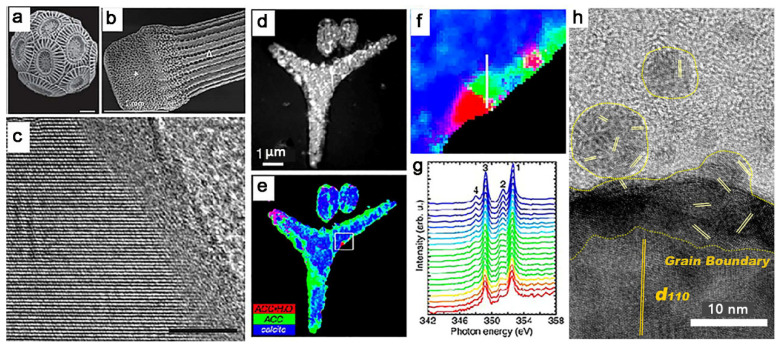
(**a**) SEM image of Emiliania huxleyi (Skeffington and Scheffel, 2018). (**b**) SEM of small spine. Reproduced with permission [81]. Copyright 2018, American Chemical Society. (**c**) HRTEM micrograph of a selected surface area of the single crystalline aragonite nacre platelets. Reproduced with permission [84]. Copyright 2005, National Academy of Science. (**d**) Component mapping in 48 h spicules, at the prism developmental stage, analyzed within 24 h of extraction from the embryo. X-ray absorption near-edge structure spectroscopy and photoelectron emission microscopy (XANES-PEEM) image of three spicules embedded in epoxy. (**e**) Red, green, and blue (RGB) map displaying the results of component mapping. The box indicates the region magnified in f. (**f**) Zoomed-in portion of the RGB map in (**e**), where each 15 nm pixel shows a different color. Pure phases are R, G, or B, whereas mixed phases are cyan, magenta, or yellow. (**g**) Sequence of 20 XANES spectra extracted from 15 nm adjacent pixels along the white line in (**f**). The color-coding is the same as that used in (**e**,**f**). Notice that the white line in f runs from the outer rim of the spicule (red), passing through orange, yellow, green, cyan, and finally blue, toward the crystalline center of the spicule. Correspondingly, moving from bottom to top across the spectra in (**g**), one can see peak-2 growth leading, and peak-4 emergence and growth lagging. Reproduced with permission [85]. Copyright 2012, National Academy of Science. (**h**) Attachment of nanocrystals onto the vaterite surface at early stages. Reproduced with permission [30]. Copyright 2016, Wiley-VCH.

**Figure 7 nanomaterials-11-00241-f007:**
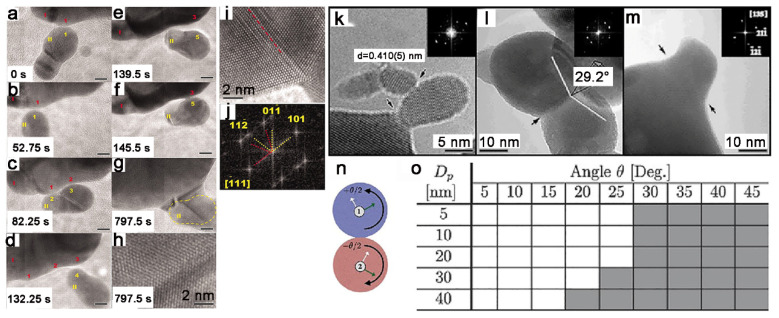
(**a**–**g**) Sequence of images showing typical dynamics of the oriented attachment (OA) process. (**h**) High-resolution image of interface in (**g**) showing twin structure (an inclined twin plane). The yellow dashed line in (**g**) shows the original boundary of the attached particle. (**i**,**j**) High-resolution in situ TEM image (**i**) and fast Fourier transform (FFT) (**j**) of an interface. The grain boundary is delineated by a dashed line in (**i**). Scale bars are 5 nm for (a) to (**g**). Reproduced with permission [94]. Copyright 2012, AAAS. (**k**) A small particle is attached to a bigger one without matching orientation. (**l**) Two particles have sintered together by building a boundary with a twin structure. (**m**) Two orientation matched particles have sintered together; the former shape is preserved. Reproduced with permission [86]. Copyright 2008, American Physical Society. (**n**) Atomistic simulation for a two-particle geometry, where two particles are rotated by an equal, but opposite angle, *θ*/2, about their common out-of-plane [001] crystal axis. (**o**) In a two-particle system, the particle diameters are Dp, the misorientation angles are angle *θ*, and the grain boundaries will not form in small misorientation angles (white cell) through rotation and will only form in large misorientation angles (gray cells). Reproduced with permission [87]. Copyright 2019, Elsevier B.V.

**Figure 8 nanomaterials-11-00241-f008:**
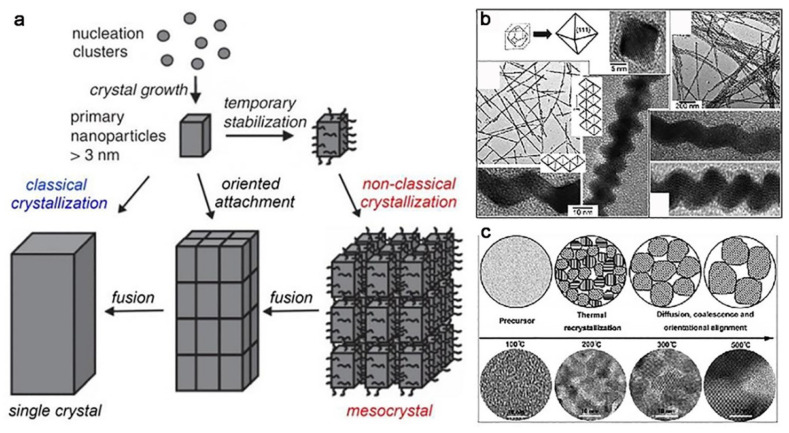
(**a**) Scheme of classical and non-classical crystallization. Reproduced with permission [105]. Copyright 2019, MDPI. (**b**) Octahedral PbSe nanocrystals grown in the presence of hexadecylamine (HDA) and oleic acid. TEM and HRTEM images of PbSe zigzag nanowires grown in the presence of HDA. Reproduced with permission [58]. Copyright 2005, American Chemical Society. (**c**) Scheme of structure and morphology transformation processes for the synthesis of Co_3_O_4_ at different temperatures. Reproduced with permission [104]. Copyright 2013, Elsevier B.V.

**Figure 9 nanomaterials-11-00241-f009:**
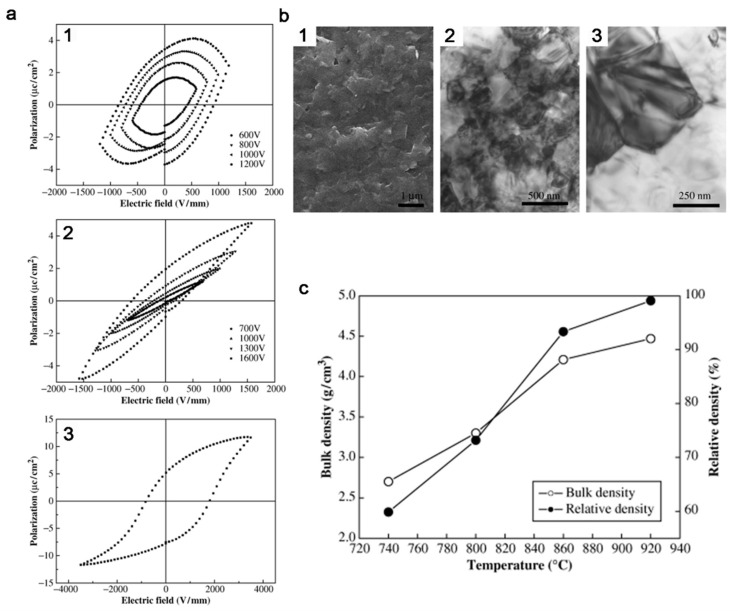
(**a**) Polarization hysteresis curves of the Na_0.5_K_0.5_NbO_3_ ceramic spark plasma sintered at 920 °C for 5 min and then annealed at 900 °C for different times in air: (1) 1 h and (2, 3) 4 h. (**b**) Microstructure of the Na_0.5_K_0.5_NbO_3_ ceramics: (1) Scanning electron microscopy micrograph and (2, 3) transmission electron microscopy micrographs. (**c**) Density change of the spark plasma sintered sample as a function of spark plasma sintering (SPS) temperature. Reproduced with permission [109]. Copyright 2005, John Wiley and Sons.

**Figure 10 nanomaterials-11-00241-f010:**
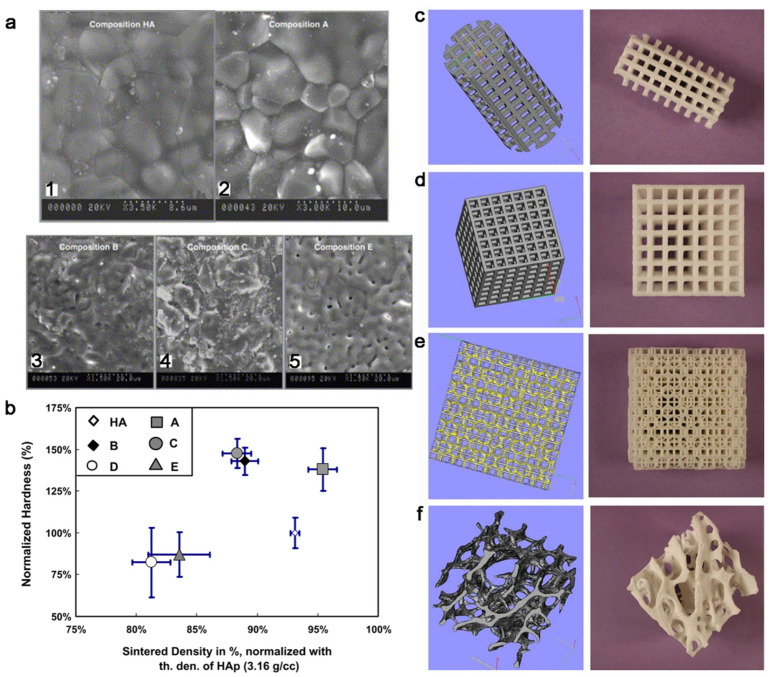
(**a**) SEM micrographs illustrating microstructures of (1) pure hydroxyapatite (HAP) and HAP with 2.5wt% of (2) 100% CaO, (3) 55% CaO + 30% P_2_O_5_ + 15% Na_2_O, (4) 30% CaO + 30% P_2_O_5_ + 40% Na_2_O, and (5) 17.2% CaO + 61.4% P_2_O_5_ + 21.4% Na_2_O. (**b**) Variation in Vickers micro hardness values of sintered compacts of different compositions as a function of their sintered densities. Reproduced with permission [120]. Copyright 2004, Elsevier. (**c**–**e**) 3D models and relative selective laser sintering (SLS) manufactured scaffold for medical application. (**f**) Micro-computer tomography of human trabecular bone tissue (left) and a mimetic tissue produced by SLS (right). Reproduced with permission [127]. Copyright 2012, Spring Nature.

**Figure 11 nanomaterials-11-00241-f011:**
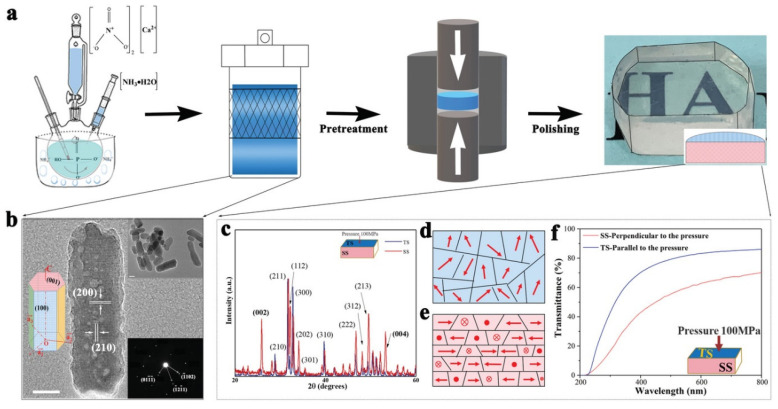
(**a**) Scheme for the preparation of hydroxyapatite (HA) optical material. (**b**) HRTEM image of the HA nanorods showing clear crystal lattice (scale bar, 10 nm). The images of mesoporous nanorods (scale bar, 20 nm) and the related selected area electron diffraction. (**c**) X-ray diffraction (XRD) patterns of the section and surface of HA ceramics prepared at 850 °C, with the increased peak intensity at (002), (004) showing that the grains have a pronounced orientation perpendicular to the pressure direction. Abridged general view (**d**,**e**) of the spatial distribution of HA grains c-axis (red arrow lines) in different pressure surfaces, exhibiting a preferable arrangement with their c-axis in the planes overlay perpendicular to the pressure direction; (**f**) The linear transmittance of HA ceramic with incident light perpendicular to and parallel to pressure directions, respectively. Reproduced with permission [7]. Copyright 2020, Wiley-VCH.

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
