# Peer review of "Construction of Inorganic Bulks through Coalescence of Particle Precursors"

_nanomaterials, 2021, doi:10.3390/nano11010241_

Round 1
Reviewer 1 Report
The paper presents a good review of methods for creating bulk materials with an ample bibliography. It is a good work but needs extensive editing for English.
I suggest to accept after the English language has been improoved.
Author Response
Comment-1: I suggest to accept after the English language has been improved. Response: Thanks for the comment. The English language of the review has been improved in revised manuscript by experts.Reviewer 2 Report
In general: The review presented could be interesting as a general introduction to coalescence phenomena, however some aspects need to be clarified.
- First at all is necessary a general revision of the English and general spelling, moreover some mistakes in the presentation (as example they have put 3.3 Crystallinity twice, instead of 3.4. Oriented attachment). I also recommend to eliminate all the etc . at the end of different examples, they are not necessary and indicate a lack of references (lines 37, 491, 544 and also others)
- From lines 66-69 the authors are talking about the misorientation between two particles, and stablish a correlation between the grain boundary energy and the angle. Can they put a number or an interval of numbers to this correlation? It is a general interval or it is completely dependent on the characteristics of the material?
- In the part 2.2 Techniques from lines 89 to 132, the introduction needs to be re written. As an introduction, the idea of sintering needs to be very clear and also its use, to produce ceramic materials. The examples used (ceria and glass ceramics) are mixing different types of sintering without a previous explanation.
- In Figures 4 and 7, the numbers in the graphics are very small and difficult to see for the reader, I recommend improving the images.
- In line 226 piezoelectrics is misspelling.
- Lines 278-282, represent a conclusion of the part 3. “The influence factors on the coalescence of particle-precursors”? If that is true is not necessary to put that here it is better to put it in the general conclusions, or alternatively indicate it.
In general a careful rewriting of the review is necessary. However it merits to be published
Author Response
Comment-1: First at all is necessary a general revision of the English and general spelling, moreover some mistakes in the presentation (as example they have put 3.3 Crystallinity twice, instead of 3.4. Oriented attachment). I also recommend to eliminate all the etc . at the end of different examples, they are not necessary and indicate a lack of references (lines 37, 491, 544 and also others). Response: Thanks for the comment and sorry for the mistyping. In the revised manuscript, we have improved our language and carefully checked the spellings. The section title in 3.4 have been revised as “Crystallographic orientation”. All the “etc.” at the end of different examples have been removed, and the potentially lacked references have also been added. Comment-2: From lines 66-69 the authors are talking about the misorientation between two particles, and stablish a correlation between the grain boundary energy and the angle. Can they put a number or an interval of numbers to this correlation? It is a general interval or it is completely dependent on the characteristics of the material? Response: Thanks for the constructive suggestion. The correlation between grain boundary energy and angle can be estimated by Read-Shockley model at low angles, which is typically around 15° for cubic materials. However, it is complex to estimate a general model of grain boundary energy at high angle. We have added the explanation in detail to show the current understanding of correlation in revised review. Please find more details in line 68-82, page 2. Comment-3: In the part 2.2 Techniques from lines 89 to 132, the introduction needs to be re written. As an introduction, the idea of sintering needs to be very clear and also its use, to produce ceramic materials. The examples used (ceria and glass ceramics) are mixing different types of sintering without a previous explanation. Response: Thanks for the comment. After careful consideration, we think the introduction of sintering technique is more appropriate in section 3.1.1 Thermal assisted interface coalescence. We have clearly introduced the idea of traditional sintering in relative section. Please see the revised texts in line 118-156, pages 3-4. Comment-4: In Figures 4 and 7, the numbers in the graphics are very small and difficult to see for the reader, I recommend improving the images. Response: Thanks for the comment. The numbers In Figures 4 and 7 have been enlarged. Comment-5: In line 226 piezoelectrics is misspelling. Response: Sorry for the misspelling. It has been revised to “piezoelectric materials”. Comment-6: Lines 278-282, represent a conclusion of the part 3. “The influence factors on the coalescence of particle-precursors”? If that is true is not necessary to put that here it is better to put it in the general conclusions, or alternatively indicate it. Response: Thanks for the comment. We agree the presence of this conclusion in original review has some misleading. In revised manuscript, we add a section title “3.1.4 Conclusion” to briefly conclude the section of 3.1. Please see more details in revised review in line 286-294, page 9.Reviewer 3 Report
1) Plagiarism- check similes and some sentences using a professional software ( see turnit-in ).
2) Scientific parts- OK, in line with the review papers but need some explanations
In introduction: Coalescence??? for every inorganic material?. Revise from aggregates ( low intermolec forces) to bulk via dangling bonds reorganizations, surface mismatch, lattice reconstruction at interfaces.
Give a comprehensive interpretation from your view related with the references
pag 2 start with line 59.... ( mechanisms).
Give a specific target for review- coalescence phenomena related to the sintering and the associated class of materials
3) Techniques - sintering via ????? it is OK but local diffusion ( interface) is temperature dependent, therefore, methods- spark, heating, microwaves need to be underlined and then discussed in section 3 in details
4) pag 11 lines 311... referring to nano coalescence- etc. It seems to be separately treated in size distribution.
In conclusion reorganize manuscript to be more comprehensive
Suggestions- in a annex - give mathematical foundations for mechanisms- it will be useful for a larger audience.
Author Response
Comment-1: Plagiarism- check similes and some sentences using a professional software ( see turnit-in ). Response: The whole review are written by our understandings and concluded by our languages. We have checked the similarities by using duplicate software. Except for technical terms, there are no similar sentences in the current version of this review. Comment-2: In introduction: Coalescence??? for every inorganic material?. Revise from aggregates ( low intermolec forces) to bulk via dangling bonds reorganizations, surface mismatch, lattice reconstruction at interfaces; Give a comprehensive interpretation from your view related with the references; pag 2 start with line 59.... ( mechanisms); Give a specific target for review- coalescence phenomena related to the sintering and the associated class of materials. Response: This is a good suggestion. This review mainly focuses on the ceramic and metal materials. We add this statement in the revised introduction so as to clarify the class of materials. Please see more details in line 40-43, page 1. We also heavily revised the statements in section 2 to have a better explanation on the mechanism of particle fusion from our understanding. Please find more detailed texts in line62-89, page 2. Comment-3: Techniques - sintering via ????? it is OK but local diffusion ( interface) is temperature dependent, therefore, methods- spark, heating, microwaves need to be underlined and then discussed in section 3 in details Response: Thanks for the suggestion. After careful consideration, we think the introduction of sintering technique is more appropriate in section 3.1.1 Thermal assisted interface coalescence. We think the revised version can eliminate the readers’ confusion for sintering and other temperature dependent techniques for particle fusion. Please find more details in line 118-156, pages 3-4. Comment-4: pag 11 lines 311... referring to nano coalescence- etc. It seems to be separately treated in size distribution; In conclusion reorganize manuscript to be more comprehensive; Suggestions- in a annex - give mathematical foundations for mechanisms- it will be useful for a larger audience. Response: Thanks for the constructive comment. This section discusses the fusion of both metal and ceramic materials. There are simulation results to conclude the correlation between particle size and melting temperature for metal particles. It indicates that smaller particle sizes have lower melting temperature. This relationship for some mineralogical ceramic is also existed. However, it lacks sufficient data to present this correlation in a mathematical formula, especially for the recent discovered clusters at sub-nanometer scale. To address the comment, we added a formula that presences the relationship between particle size of metal and melting temperature in line 296-310, page 9.Round 2
Reviewer 1 Report
The paper can be published with the corrections made
Reviewer 2 Report
The authors have satisfactory answered the questions/changes I have asked for. I recommend the publication
Reviewer 3 Report
Well done